# Influence of Transition Metals (Cu and Co) on the Carbon-Coated Nickel Sulfide Used as Positive Electrode Material in Hybrid Supercapacitor Device



**Souvik Ghosh** [1,2], **Aparna Paul** [1,2], **Prakas Samanta** [1,2], **Bhau Landge** [3], **Sanjib Kumar Mandal** [4], **Sangam Sinha** [3], **Gour Gopal Dutta** [4], **Naresh Chandra Murmu** [1,2] **and Tapas Kuila** [1,2,*]

[1] Surface Engineering & Tribology, Council of Scientific and Industrial Research-Central Mechanical Engineering Research Institute, Durgapur 713209, India; sgkssm92@gmail.com (S.G.); aparnainspire@gmail.com (A.P.); samantaprakas2@gmail.com (P.S.); murmu@cmeri.res.in (N.C.M.)

[2] Academy of Scientific and Innovative Research (AcSIR), AcSIR Headquarters CSIR-HRDC Campus, Postal Staff College Area, Sector 19, Kamla Nehru Nagar, Ghaziabad 201002, India

[3] Vehicles Research and Development Establishment (VRDE), Defence R &D Organization (DRDO), Vahannagar, Ahmednagar 414006, India; balandge@vrde.drdo.in (B.L.); director@vrde.drdo.in (S.S.)

[4] Jagadish Chandra Bose Centre of Advanced Technology, Defence R &D Organization (DRDO), IIT Kharagpur Research Park, New Town, Kolkata 700160, India; sanjibkumar.mandal@jcbcat.drdo.in (S.K.M.); director@jcbcat.drdo.in (G.G.D.)

\* Correspondence: tkuila@gmail.com or t_kuila@cmeri.res.in

**Abstract:** Herein, a facile, environment-friendly and cost-effective approach was followed for the preparation of metal sulfide-based supercapacitor electrodes. The effect of transition metal interrogation on the morphology and electrochemical performance of carbon-coated nickel sulfide composite electrode was investigated. Physicochemical characterization showed that the enhancement in electrical conductivity and electrochemical reaction sites with the introduction of copper (Cu) and cobalt (Co) was due to the variation in morphology. Fast ionic transformation and improvement in the number of redox active sites might improve the supercapacitor performance. The electrochemical experiment showed that the NCoSC electrode exhibited the highest capacitance value of ~760 F $g^{-1}$ at 2 A $g^{-1}$ current density as compared to the NCuSC and NSC electrodes. Therefore, a hybrid supercapacitor (HSC) device was fabricated by using NCoSC as the positive electrode and thermally reduced graphene oxide (TRGO) as the negative electrode. The fabricated device demonstrated maximum energy density of ~38.8 Wh $Kg^{-1}$ and power density of 9.8 kW $Kg^{-1}$. The HSC device also showed ~89.5% retention in specific capacitance after 10,000 charge–discharge cycles at 12 A $g^{-1}$ current density. So, the tuning of electronic and physical properties by the introduction of Cu and Co on nickel sulfide improved the supercapacitor performance.

**Keywords:** electrode materials; band gap energy; specific capacitance; hybrid supercapacitor; EIS

## 1. Introduction

In recent years, environmental pollution and global warming are the major concerns due to the increasing energy demand for the industrialization and economical growth. Among the various energy storage and conversation devices, supercapacitors have drawn extensive attention due to its long cycle stability, ultrafast charging rate, and eco-friendliness. However, the low energy density value limits its wide range of practical applications. Therefore, the improvement of energy density of supercapacitors is the prime focus of research nowadays without affecting the other electrochemical properties [1–4]. Electrode materials play a key role on the performance of the supercapacitor devices. Therefore, in order to improve the electrochemical performance of supercapacitors, it is necessary to engineer the electrode materials. Supercapacitors store charge by two mechanisms. In the first mechanism, electrolyte double layer capacitance (EDLC),the electrolyte ions are

adsorbed on the electrode materials and form a Helmholtz-like layer, which mainly supply high-power density. The second is pseudocapacitance pathway where Faradic type redox reaction occurs, which supply high-energy density of supercapacitor [5–7]. Transition metal chalcogenides (TMC) with variable oxidation states have shown good electrochemical energy-storage properties. Among the various TMCs, nickel (Ni), cobalt (Co), and cupper (Cu)-based materials have been studied extensively as supercapacitor electrode due to their high natural abundance, large number of oxidation states, and high content of electro active sites. Ternary metal sulfides such as $NiCo_2S_4$, $NiCoS_2$, $NiCu_2S_4$, $NiMnS_4$ etc., have been reported to exhibit higher electrochemical properties with improved electrical conductivity than single Ni- or Co- sulfides [5,6,8,9]. The electrical conductivity is an important parameter, which affects the electrochemical activity and kinetic properties of the supercapacitor electrode materials. Highly conductive material facilitates easy and fast electron transport. Therefore, it is compulsory to reduce the resistance properties of the material and improve the electrochemical performance of the supercapacitor. Interrogation of transition metal atoms (such as Cu and Co) into the TMCs to form a bimetallic composite is an emerging approach to overcome the electrical conductivity issues [10–12]. In addition, formation of three-dimensional (3D) composite materials is also a desirable approach to enhance the electrochemical performance of supercapacitor. 3D electrode materials facilitate easy electron transfer, effective accessibility, and good contact between the active electrode materials and electrolyte ions [10,11,13]. Lower electronegativity of the sulfur as compared to the oxides facilitates the electron transfer process and improves the conductivity of the electrode materials. However, the agglomeration tendency of nickel sulfide-based electrode materials reduces electrochemically active sites, resulting in poor electrochemical properties like inferior rate capability and stability. On the other hand, carbonaceous materials reduce the agglomeration of the TMC-based materials resulting larger electroactive surface area and improved electrical conductivity. In another approach, a thin layer coating of carbon on the TMC materials further increased the structural stability of the active electrode materials during long charge-discharge cycles [14–18]. A few studies showed the use of only nickel/copper/cobalt chalcogenide-based or nickel chalcogenide-based bi-metallic composite as electrode materials in supercapacitor devices [16–18]. Tavakoli et al. reported the use of yolk-shell-like $CuCo_2Se_4$ microspheres as supercapacitor electrode material. The hollow yolk-shell-like structure decreased the mass/charge transfer distance and increased the rate capability with power density of the supercapacitor. It showed high specific capacitance of ~512 F $g^{-1}$ at 1 A $g^{-1}$ current density [1]. Ma et al. showed that the intertwined fiber structure is advantageous for supercapacitor applications. It was found that the $NiCo_2S_4$-embedded carbon fiber displayed improved capacitive properties compared to the pure $NiCo_2S_4$ [5]. X. sun and co-workers prepared $CNT@NiMn_2O_4$ core-shell nano-structured composite through microwave-assisted hydrothermal techniques [8]. In comparison to the pure $NiMn_2O_4$, the $CNT@NiMn_2O_4$ core-shell structure exhibited better supercapacitor performance due to the improved electrolytes transportation. Wan et al. synthesized anisotropic carbon from the wood derivative and formed a composite with copper oxide for supercapacitor application. The abundant anisotropic channels present in carbon facilitated rapid transportation of electrolytes and reduced the resistance of the electrode material. The electrode material exhibited maximum specific capacitance of 694.8 F $g^{-1}$ at 0.5 A $g^{-1}$ current density [9]. Therefore, it is seen that the porous carbon atoms can reduce the resistance and increase the rate capability and power density of the supercapacitor device. However, the effect of the addition of transition metal atoms in nickel chalcogenide-based electrodes on the morphology, electrical properties, and charge storage performance has not been reported so far.

Based on the above considerations, carbon-coated bi-metallic nickel sulfide-based composite was prepared by a one-step hydrothermal technique for the development of high-performance electrode materials for supercapacitor. In this study, introduction of transition metals (Co and Cu) into the carbon-coated nickel sulfide is expected to improve the performance of the supercapacitor electrode. The influence of the physical

and electrochemical properties of electrode materials for the interrogation of two transition metals separately was systematically studied.

## 2. Materials and Methods

### 2.1. Materials

Nickel chloride hexa hydrate ($NiCl_2.6H_2O$), cobalt chloride hexa hydrate ($CoCl_2.6H_2O$), copper nitrate, sodium thiosulphate di-hydrate ($Na_2S_2O_3.2H_2O$), and N, N-dimethyl formamide (N,N-DMF) were purchased from merk Specialities Private Limited, Vikhroli East, Mumbai 400 079, India. Ascorbic acid and glucose were purchased from Sigma-Aldrich (Missouri, United States, USA). Polyvinylidine fluoride (PVDF) as binder and carbon black as conducting agent (EC-600JD, purity: >95%) were obtained from Akzo Nobel Amides Co., Ltd., 1070 - 2 Hwangsung-dong Gyeongju, 780952, Korea Rep. Nickel foam (NF) was obtained from GELON LIB CO., Ltd., songshanhu district, dongguan City, Guangdong province 723808, China.

### 2.2. Preparation of the Electrode Materials

The electrode materials were synthesized by mixing 1 mM $NiCl_2.6H_2O$ and 2 mM $Na_2S_2S_3.2H_2O$ in 20 mL DI water in a round bottom (RB) flux and in another RB flux 10 mM glucose and 0.1 mM ascorbic acid were mixed in 10 mL DI water, stirred thoroughly for ~30 min. Glucose acted as carbon source and ascorbic acid acted as the reducing agent. Then, the two solutions were mixed and stirred again for ~30 min. The whole mixture was transferred to a 100 mL autoclave reactor inside a preheated oven at 180 °C for 12 h. Then the autoclave reactor was cooled down to room temperature naturally. The final material (designated as NSC) was obtained by vacuum filtration with DI water and ethanol followed by drying inside a vacuum oven overnight. Another two composites were prepared following the same procedure using 2 mM of $CoCl_2.6H_2O$ and $Cu(NO_3)_2$ separately with nickel and sulfide precursors, designated as NCoSC and NCuSC, respectively.

Graphene oxide (GO) was prepared by modified Hummers method [12]. Thermally reduced GO (TRGO) was prepared by thermal annealing of ~100 mg of freeze-dried GO inside a muffle furnace at 500 °C for 30 min. The formation of TRGO was confirmed by FT-IR and Raman spectra studies. The FT-IR spectra as shown in Figure S1a demonstrated that the peak intensity of oxygenated groups at 1622, 1400, 1118, and 1030 $cm^{-1}$ regions decreased remarkably after the thermal treatment of GO confirming the formation of TRGO by the thermal treatment of GO. Raman spectra study (Figure S1b) also showed higher G band (1576 $cm^{-1}$) intensity compared to D band (1360 $cm^{-1}$) confirming the lower disorder in TRGO and larger number of $sp^2$ hybridized carbon atoms. Appearance of a sharp 2D band at ~2765 $cm^{-1}$ region confirmed the formation of a few layers of TRGO.

### 2.3. Structural Characterization of the Materials

The morphology of the synthesized electrode materials was evaluated by field emission scanning electron microscopy (FE-SEM). The crystallinity and phase of the materials were determined by powder X-ray diffraction (Rigaku Mini Flex 600, Japan) using Cu K$\alpha$ radiation with $\lambda$ = 1.5418 Å (at diffraction angles (2θ) ranging from 10 to 80° at the scan rate of 5° $min^{-1}$). Perkin Elmer RXI FT-IR in the frequency range of 4000–700 $cm^{-1}$ was used to obtain the Fourier transform infrared (FT-IR) spectra of the electrode materials. The optical properties and the direct band gap energy of the electrodes were calculated through ultra-violet spectroscopy using Shimadzu UV-1800 spectrophotometer. Raman spectra of the composite materials were examined on a Witec alpha300 R instrument using Laser wavelength of 532 nm. The electrical conductivity of the electrode materials were measured by a KEITHLEY delta system consisting of a Nanovoltmeter (model: 2182A) with an AC and DC current source (model: 6221) using a four-probe set up. Equation (1) was used to calculate the electrical conductivity of the materials.

$$\text{Electrical conductivity} = 1/4.53 \times R \times d \tag{1}$$

where, "R"is the resistance of the materials and "d"is the thickness of the sample pellets. The current range during the measurement was $-20 \times 10^{-6}$ to $20 \times 10^{-6}$ Amp and the voltage sweep was fixed at 100 mV s$^{-1}$ for all the measurements.

### 2.4. Electrochemical Characterization

The cyclic voltammetry (CV), galvanostatic charge-discharge (GCD) and electrochemical impedance spectroscopy (EIS) were recorded in three-electrode configuration with PARSTAT 3000 (Princeton Applied Research, USA) electrochemical workstation. About 6M KOH was used as electrolyte inelectrochemical experiment, in which Pt wire and AgCl/Ag saturated with KCl were used as counter and reference electrode, respectively. The working electrode was prepared by mixing active electrode materials, carbon black, and PVDF at 85:10:5 ratios in N, N-DMF solvent, and the obtained slurry. Then the prepared slurry was drop casted on NF and dried inside a vacuum oven at 60 °C for 12 h.

### 2.5. Calculation

The specific capacitance of the electrodes and HSC device was calculated using Equation (2),

$$SCs = i \times \Delta t / m \times \Delta V \tag{2}$$

where, "i" denotes the constant discharge current and "$\Delta t$"is the full discharging time. "m"is the mass of the corresponding active material and "$\Delta V$"is the working potential window.

The charge balance of the positive and negative electrode materials were calculated by using the following Equation (3):

$$q = m \times \Delta V \times SCs \tag{3}$$

After charge balancing, the mass loading ratio of electrodes in HSC device can be calculated using Equation (4):

$$m^+/m^- = SCs^- \times \Delta V^- / SCs^+ \times V^+ \tag{4}$$

where, "m"is the mass of the electrode material, charge of the electrode is "q"and "$\Delta V$"is the working potential window. "$\Delta t$"is the discharge time from GCD curve and "SCs"is the specific capacitance of the electrode.

The energy density (ED) and power density (PD) of the fabricated HSC device was determined according to the following equation, respectively (Equations (5) and (6)),

$$ED = SCs \times (\Delta V)^2 / 2 \times 3.6 \tag{5}$$

$$PD = ED / \Delta t \tag{6}$$

where, the symbols refer to the inner meaning.

## 3. Results and Discussion

### 3.1. PXRD Pattern Analysis

The crystallinity and phase of the synthesized materials were characterized by powder X-ray diffraction (PXRD) analysis as shown in Figure 1a. A series of sharp peaks indicated the crystalline nature of the obtained electrode materials. The diffraction peaks appeared at 2θ values of ~27.7, 32.1, 35.8, 39.5, 45.9, 54.6, 59.7, and 62.3° and can be well indexed to the (111), (200), (210), (211), (220), (311), (021), and (321) crystalline planes of NiS$_2$ (JCPDS card file No.11–99), respectively [19,20]. Absence of any other peaks indicated good crystallinity and purity of the electrode material. In addition to the NiS$_2$ peaks, the PXRD pattern of NCuSC composite showed additional peaks located at 2θ of ~27.00, 29.2 33.2, 44.6, 47.9, 53.1° corresponding to the (101), (102), (006), (106), (110), and (108) crystalline planes of CuS (JCPDS card file No. 3-424). However, all the peaks related to CuS and NiS$_2$ slightly shifted from the respective peaks present in pure materials. This observation

suggested the strong attraction of CuS and $NiS_2$ particles and the formation of $NiCuS_2$ composite materials [21,22]. On the other hand, the PXRD pattern of NCoSC electrode material is well indexed to the formation of $NiCo_2S_4$ composite. The characteristic peaks at 2θ = 27.3, 31.6, 34.7, 38.9, 45.2, 53.6, 56.1, 58.5, and 61.4° were owing to the crystal planes of (220), (311), (222), (400), (422), (511), (440), (531), (620) of $NiCo_2S_4$, respectively. The PXRD pattern showed that in comparison to the $NiS_2$, the peaks of NCoSC electrode aremore intense, indicating the homogeneous distribution of Co in $NiS_2$ and $NiCoS_2$. In addition to the above peaks, a small peak also appeared at 2θ ≈ 18°, which corroborated well with the characteristic (111) plane of $NiCo_2S_4$ [5,13,17]. A broad hump at 2θ ≈ 23 to 26° was detected in the PXRD pattern of all the samples due to the (002) plane of coated carbon. Appearance of this broad hump suggested the amorphous and graphitic nature of the coated carbons [12,17].

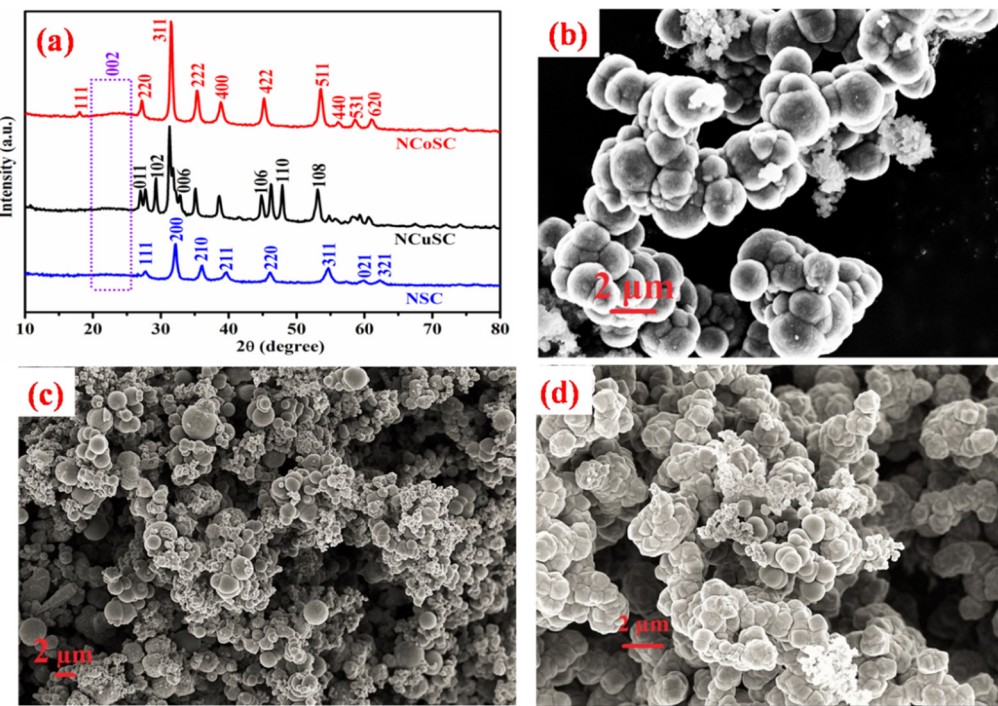

**Figure 1.** (**a**) PXRD pattern of NSC, NCuSC, and NCoSC electrodes, (**b–d**) FESEM images of NSC, NCuSC, and NCoSC electrodes, respectively.

### 3.2. FE-SEM and HR-TEM Images Analysis

FE-SEM images shown in Figure 1b–d were recorded to analyze the surface morphology of the synthesized electrode materials. The FE-SEM images showed that all the electrode materials formed spherical 3D morphology. The FE-SEM image of NSC as shown in Figure 1b exhibited that the particles were spherical in nature and formed a densely stacked heterogeneous structure. The growth of such type of large and stacked particles may be infeasible for ion transport during the electrochemical reactions [23]. The size of $NiS_2$@C particles was relatively large as compared to the NCuSC (Figure 1c) and NCoSC (Figure 1d). On the contrary, the FE-SEM images of NCuSC composite electrode showed that the $NiCuS_2$@C particles were uniformly distributed. However, the particle sizes were not consistent. The FE-SEM image of NCoSC composite electrode (Figure 1d) exhibited unique size, well distribution, and interconnected particles. This type of interconnected network of the particles may increase the number of electrochemical active sites and improve the electrochemical properties of the supercapacitor device [24,25]. Therefore, the FE-SEM images showed that the introduction of other transition metals (Cu and Co) with $NiS_2$@C resulted in uniform growth of NCuSC and NCoSC particles with smaller size, which may increase the active space for electrolytes and enhance the diffusion of the electrolytes ion.

The HR-TEM image of NCoSC electrode was carried out to characterize the carbon-coated nickel sulfide electrodes. The HR-TEM image of NCoSC electrode (Figure 2a) showed network-like interconnected structure of the particles. The d-spacing of the NCoSC electrode was calculated in the A and B region as shown in Figure 2a. The fast Fourier transformation (FFT) followed by inverse FFT (IFFT) was carried out in the selected area. The d-spacing obtained from the IFFT images from region A (Figure S2a) was found to be 0.23 nm confirming the presence of (400) plane of $NiCo_2S_4$. The d-spacing of 0.366 nm in region B indicated the (002) plane of coated carbon (Figure S2b). The incomplete destruction of the crystallographic features of (002) plane of coated carbon suggested the presence of amorphous nature of the coated carbons. The HR-TEM images confirmed the carbon coating on the $NiCo_2S_4$ particles and formation of NCoSC composite electrode materials. The (400) plane of $NiCo_2S_4$ and (002) plane of carbon as evidenced from the HR-TEM image analysis matched well with the PXRD pattern of NCoSC [5,9,12,21,22].

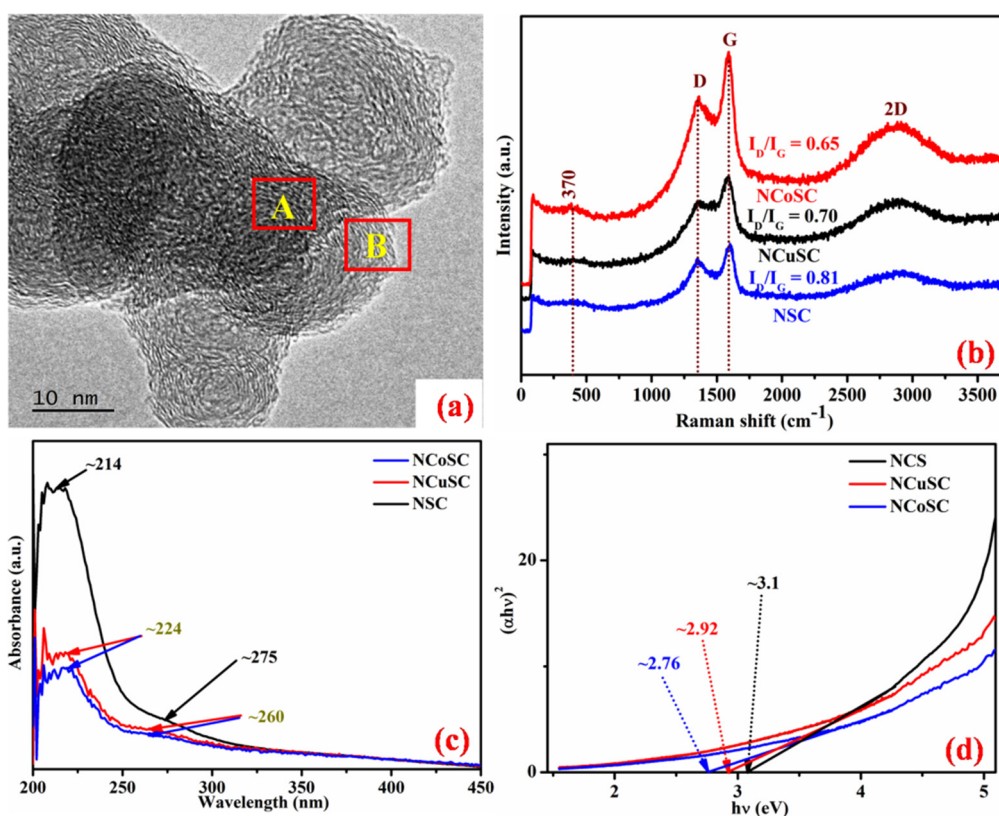

**Figure 2.** (**a**) HR—TEM image of NCoSC (**b**) Raman spectra (**c**) UV—vis spectra and (**d**) $(\alpha h\nu)^2$ vs. $h\nu$ plot of NSC, NCuSC, and NCoSC electrodes.

*3.3. Raman Spectra Analysis*

The Raman spectra of NSC, NCuSC, and NCoSC electrode materials are shown in Figure 2b. The Raman peaks at 1356 and 1578 $cm^{-1}$ correspond to the D and G modes, respectively. The higher G band intensity compared to the D band in all the composite materials suggested that the coated carbon atoms were more $sp^2$ hybridized and the disorder nature of the carbon atoms was smaller. The $I_D/I_G$ ratio decreased from NSC (0.81) to NCuSC (0.70) to NCoSC (0.65) electrodes periodically implying the decrease in disorder structure of coated carbon and the synergistic effect of metal sulfide with carbon. A broad 2D band present in all the electrode materials at 2850 $cm^{-1}$ region was ascribed to the graphitic nature of the coated carbon. The 2D band intensity increased gradually from NSC to NCuSC to NCoSC electrode. It is suggested that the coated carbon on metal sulfide formed a layered structure similar to graphite [3,5,8,19].

### 3.4. FT-IR Spectra Analysis

Figure S3 represents the FT-IR spectra of the synthesized composite electrode materials. A broad –OH absorption band at ~3440 cm$^{-1}$ appeared in all the three electrode materials. The othertwo peaks at ~1630 and 1400 cm$^{-1}$ regions originated due to the stretching vibration of –C=O and conjugated C=C. The absorption peak at 1100 cm$^{-1}$ corresponded to the deformation vibration of the –C-O single bond [26,27]. The peak intensity of –C=O stretching vibration increased in the order of NSC < NCuSC < NCoSC, which indicated that the amount of carbon contain was well maintained in the NCuSC and NCoSC electrodes. In addition, another peak appeared at 655 cm$^{-1}$ region for all the samples which was possibly due to the Ni-S-Ni bending vibration. NCuSC and NCoSC electrodes showed another peak at 784 cm$^{-1}$ due to the stretching vibration of Cu-S/Co-S [19,28]. Hence, the FT-IR spectra further validated the formation of NSC, NCuSC, and NCoSC composite electrode materials.

### 3.5. UV-Visible Spectroscopy Analysis

Electrical properties were investigated by UV-visible absorption spectroscopy and used to measure the electrochemical band gap energy. Figure 2c represents the UV-visible spectroscopy of NCS, NCuSC, and NCoSC electrodes. The spectrum of all the electrodes (Figure 2b) showed two peaks at different regions of 210 to 230 nm and 260 to 280 nm. A broad absorbance band at ~275 nm region of NSC electrode implied the presence of metal sulfides. This broad peak slightly shifted to lower wavelength in NCuSC and NCoSC electrodes (~260 nm) due to the introduction of Cu and Co in carbon-coated nickel sulfide compounds [29]. Another peak appeared at ~224 nm of NCuSC and NCoSC electrodes correspond to the stretching vibration of –C=O groups of coated carbon. This peak slightly shifted to the lower wavelength of ~214 nm in NSC electrode. Band gap was measured from the UV-vis absorption spectrum to evaluate the electronic properties of the electrode materials. The optical band energy ($E_g$) can be measured by the Tauc's plot of the following equation,

$$(\alpha h\upsilon) = K\,(h\upsilon - E_g)^n \tag{7}$$

where, "$\alpha$"is the absorption coefficient, "$h\upsilon$"is the photon energy, "K"is the constant relative to the materials, and "$E_g$"is the band gap energy. The integer "n"is $\frac{1}{2}$ for the direct band gap and 2 for indirect band gap. The direct band gap of the synthesized electrode materials was calculated by plotting the graph [$(\alpha h\upsilon)^2$ vs. $h\upsilon$] (Equation (7)), and the X-axis intercept to zero gives the band gap energy value. Figure 2d shows that the band gap energy of the NCS (~3.1 eV) electrode was higher compared to the NCuSC (~2.92 eV) and NCoSC (~2.76 eV) electrodes. The band gap energy indicated that the electrical conductivity increased with the incorporation of transition metals in carbon-coated nickel sulfide [12,30]. This observation was also confirmed by measuring the electrical conductivity of the materials using four probe techniques.

### 3.6. Electrical Conductivity Analysis

The electrical conductivity of all the synthesized electrode materials was measured by four probe technique. The electrical conductivity of NSC, NCuSC, and NCoSC electrode materials were calculated as 2.74, 36.4, and 32.2 S m$^{-1}$, respectively. The electrical conductivity values also suggested that the materials became more conductive with the incorporation of transition metal in carbon-coated nickel sulfide. The slightly higher conductivity of NCuSC than NCoSC is due to the presence of easily donating outer shell i.e., $4s^1$ electron of copper compared to more stable outer shell $4s^2$ electron of cobalt. The increase in electrical conductivity suggested the semi-conductive nature of the NCuSC and NCoSC composite electrode materials compared to the NSC composite electrode. The electrical conductivity values agreed well with the band gap energy of the electrode materials [31,32].

### 3.7. Electrochemical Characterization of Electrodes

The electrochemical properties of the electrode materials were determined by CV, GCD, and EIS measurements in three-electrode configurations. Figure 3a displays the comparative CV curves of NSC, NCuSC, and NCoSC at 30 mV s$^{-1}$ scan rate over the working potential window of −0.1 to 0.4 V. The CV curves showed that the integrated area of NCoSC was larger than the other two implying the best electrochemical performance of the NCoSC electrode. The CV curves also showed a pair of redox peaks in all the electrodes indicating that the main charge storage characteristics were dominated by the reversible Faradic redox reaction of metal sulfides [2,4,33–35]. CV curves also suggested that the NCoSC and NCuSC electrodes showed pure battery-like behavior. In contrary, the capacitive property of NSC electrode was governed by the battery-like charge storage process as well as the surface-controlled pseudocapacitive charge storage mechanism. Figure 3b is the CV curves of NCoSC electrode at different scan rates of 10 to 200 mV s$^{-1}$. The two pairs of redox peaks that appeared in the CV curves were originated due to the redox reaction of Ni$^{+2}$/Ni$^{+3}$ and Co$^{+2}$/Co$^{+3}$ transitions associated with OH$^{-}$. The CV curves also showed that with increasing the scan rates the peaks position did not change suggesting the high reversibility, good rate capability, and low resistivity of the electrode material. The OH$^{-}$ ions took part in the electrochemical reaction of transition metal sulfides similar to the transition metal oxides. The redox reactions in the aqueous KOH electrolytes were based on the following Equations (8)–(10) [24,36,37]:

$$NiS + OH^{-} = NiSOH + e^{-} \tag{8}$$

$$CoS + OH^{-} = CoSOH + e^{-} \tag{9}$$

$$CoSOH + OH^{-} = CoSO + H_2O + e^{-} \tag{10}$$

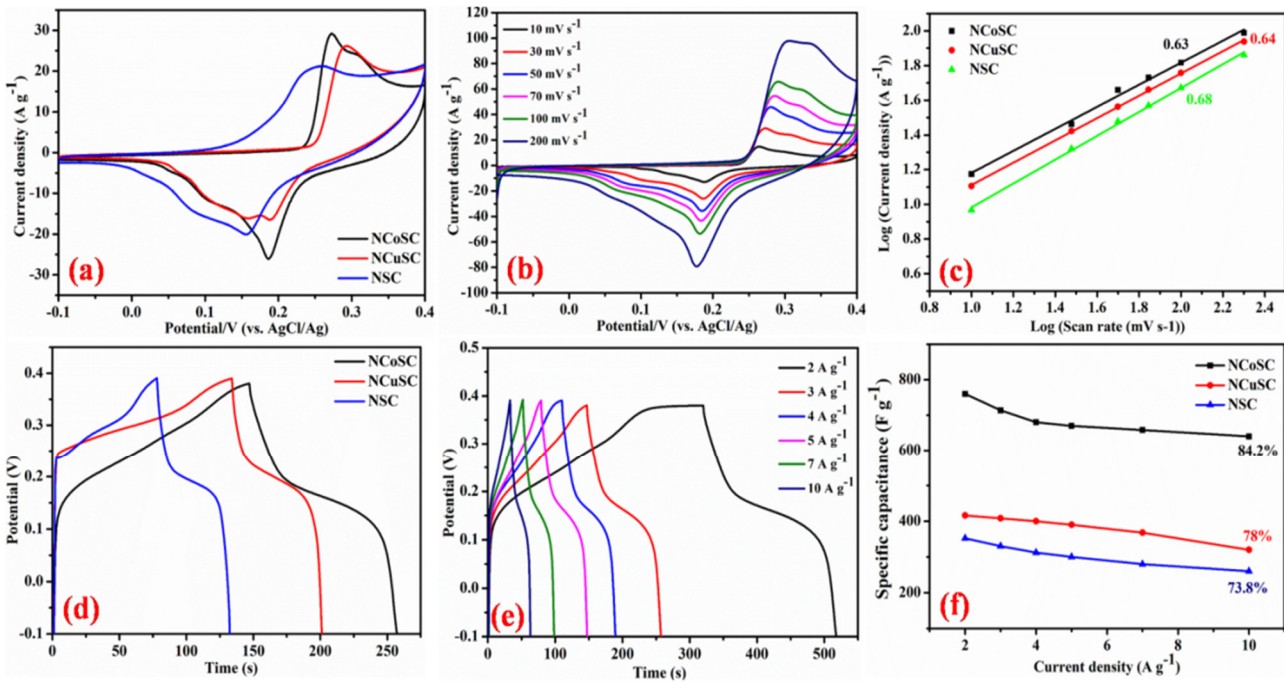

**Figure 3.** (**a**) Comparable CV curves of NSC, NCuSC, and NCoSC electrodes; (**b**) CV curves of NCoSC electrode at different scan rates; (**c**) Log (i) vs. Log (v) plots of NSC, NCuSC, and NCoSC electrodes; (**d**) comparable GCD plots of NSC, NCuSC, and NCoSC electrodes; (**e**) GCD profile of NCoSC electrode at different current density; (**f**) comparable specific capacitance vs. current density curves of NSC, NCuSC, and NCoSC electrodes.

　　　The CV curves of NCuSC and NCS electrodes at different current density are shown in Figure S4a,b. Those CV curves indicated that the peak positions did not change significantly with increasing the scan rates.

　　　The charge storage mechanism of the electrodes was evaluated by measuring the anodic peak current according to the scan rate. Whether the charge storage is capacitive controlled or diffusion controlled can be measured by plotting the log (i) vs. log (V) following the power law relationship (Equations (11)–(12)) [6,12,19,38]:

$$i = aV^b \qquad\qquad\qquad (11)$$

$$\log (i) = \log (a) + b \log (V) \qquad\qquad\qquad (12)$$

where, i and V signify the response peak current and scan rate, respectively. The other two terms "a" and "b" are variable parameters and the "b" value can be calculated from the slop of the straight line. When the value of "b" is 0.5, it indicates the diffusion-controlled charge storage process and b = 1 signifies the surface capacitive-controlled charge storage process. The "b" values of the anodic peak were calculated to be 0.63, 0.64, and 0.68 (Figure 3c) of NCoSC, NCuSC, and NSC electrodes, respectively. The calculated value of "b" indicated that the redox process was controlled by the diffusion redox process when the transition metals were introduced in carbon-coated nickel sulfides [4,6,38]. The capacitance of the electrode materials was measured from the discharge time of the galvanostatic charge–discharge profiles. Figures 3d and S4c show the comparative GCD curves of the NSC, NCuSC, and NCoSC electrodes at constant current density of 3A g$^{-1}$. The GCD curves of all the three electrodes were non-linear and distinct voltage plateaus were observed indicating the typical Faradic type reaction. As expected from the CV curves, the NCoSC electrode showed high discharge time as compared to the NCuSC and NSC electrodes, indicating higher capacitance of the NCoSC electrode. The specific capacitance values were calculated as ~714, 408, and 318 F g$^{-1}$ for NCoSC, NCuSC, and NSC electrodes, respectively at 3 A g$^{-1}$ constant current density. Figure 3e shows the GCD curves of the NCoSC electrode at different current density of 2 to 10 A g$^{-1}$ and the GCD profile retained their shape at high current density. The GCD curve of the NCuSC and NSC at different current density (2–10 A g$^{-1}$) electrode were also demonstrated in Figure S4d,e, respectively. The FE-SEM image of NCoSC electrode showed the interconnected network-like morphology, which provided the percolated conducting pathway and it showed higher redox kinetics. The NCoSC electrode also showed large number of surface expose redox active sites and facile redox kinetics. The change in specific capacitance with current density is shown in Figure 3f. The retention in specific capacitance of NCoSC electrode (84.2%) was better as compared to the NCuSC (78%) and (73.8%) electrodes at higher current density of 10 A g$^{-1}$. Higher retention of capacitance of NCoSC is attributed to the synergistic effect between the transition metals carbon coating. Improvement in electrical conductivity is attributed to the interconnected network structure of NCoSC electrode, contributing to achieving higher electrochemical properties [4–6,12,32,38].

　　　EIS is a principal method to measure the internal resistance and ion diffusion properties of the materials. Figure 4a represents the Nyquist plot of the NSC, NCuSC, and NCoSC electrodes. The Nyquist plot was obtained to measure the impedance of the electrode materials in the frequency range of 0.1 to 10,000 Hz at open circuit potential with AC perturbation of 10 mV. The Nyquist plots were simulated by Z-View software with the help of equivalent Randle circuit (in set of Figure 4a). The Randle circuit contains four components i.e., uncompensated resistance ($R_s$), constant phase element (CPE), charge-transfer resistance ($R_{ct}$), and Warburg (diffusion) resistance ($W_o$). At high frequency region, the intercept at the X-axis of the Nyquist plot represents the $R_s$ and is governed by the contact resistance of the electrode materials with current collector and electrolytes [8,19,38]. The semi-circle along the X-axis represents the charge-transfer resistance of the electrode material and can be obtained at mid to high frequency region, mainly the $R_{ct}$. At lower frequency region, a vertical line parallel to Y-axis represents the $W_o$ and it is generated due to the electrolyte

ion diffusion. The CPE is dependent on frequency and it represents the deviation of the capacitive nature of the electrode materials from the ideal capacitive behavior. The CPE can be divided into two physical constituents, one is the phase element exponent (CPE-P) which indicates the nature of the charge transfer process at high frequency region and the other is the capacitance extent (CPE-T) that characterizes the double layer capacitance of active electrode materials.

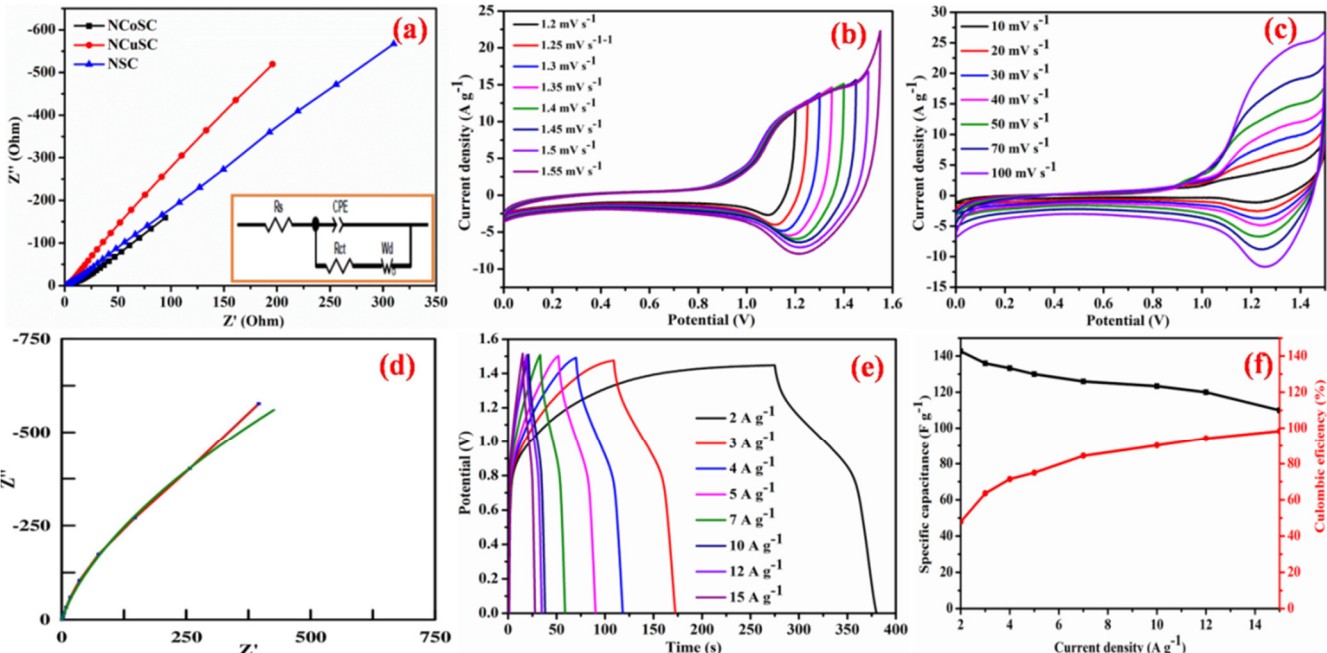

**Figure 4.** (**a**) EIS spectra of NSC, NCuSC, and NCoSC electrodes and Randle circuit (inset); (**b**) CV curves of HSC device at different working potential window; (**c**) CV curves HSC device; (**d**) fitted EIS spectra HSC device; (**e**) GCD plot at different current density HSC device; and (**f**) specific capacitance and columbic efficiency vs. current density plot of fabricated HSC device.

The Z-view fitted curves of the NSC, NCuSC, NCoSC electrodes are shown in Figure S5a–c, respectively and the fitted results are summarized in Table 1. From the fitted data, it is seen that the Rs values of the three electrodes are lower and almost same indicating the contact resistance between the electrode and current collector and nature of the electrolyte resistance [31,32,38]. A small semi-circle is observed in Figure S5d for all the electrodes suggesting the lower $R_{ct}$ value. The fitted data showed that the $R_{ct}$ value gradually decreased from NSC (30 Ω) to NCuSC (15Ω) to NCoSC (10 Ω) electrodes. The lower $R_{ct}$ value of NCoSC electrode indicated good electrical conductivity, low charge transfer resistance, and favorable ionic transport in the electrolyte medium. The electrolyte ion diffusion behavior of the electrodes was obtained from the $W_o$ value of the fitted EIS data. The Warburg element ($W_o$) can be divided into three components of Warburg resistance (W-R), diffusion time constant (W-T) and the phase exponent (W-P). W-T represents how fast electrolyte ions are diffused and W-P represents the capacitive diffusion nature of the electrolytes. The fitted data demonstrated Warburg resistance of the NSC and NCuSC electrodes were almost same and decreased in the NCoSC electrode. The lower $W_o$ value of the NCoSC electrode signified facile and shorter ion diffusion path to the active electrode material with more ideal capacitive behavior compared to the other two electrodes [2–6,31,38]. Electrochemical measurement suggested that the NCoSC material showed superior performance as compared to the NSC and NCuSC electrodes. The electrochemical measurement also suggested improved supercapacitor performance with the incorporation of Cu and Co separately with carbon-coated nickel sulfides.

**Table 1.** Fitted values of $R_s$, $R_{ct}$, CPE-T, CPE-R, W–R, W–T, W–P from the equivalent circuit corresponded to Figure S2a–c.

| Composite | $R_s$ ($\Omega$) | CPE-T ($S^{-n}$) | CPE-P (n) | $R_{ct}$ ($\Omega$) | W-R ($\Omega$) | W-T (s) | W-P |
|---|---|---|---|---|---|---|---|
| NSC | 1.2 | 0.0015 | 0.75 | 30 | 120 | 0.045 | 0.32 |
| NCuSC | 1.4 | 0.0015 | 0.85 | 15 | 118 | 0.055 | 0.35 |
| NCoSC | 1.5 | 0.0035 | 0.65 | 10 | 90 | 0.26 | 0.37 |

*3.8. Electrochemical Characterization of the Fabricated Hybrid Supercapacitor (HSC) Device*

A hybrid supercapacitor (HSC) device was fabricated by NCoSC electrode as positive electrode and TRGO as negative electrode (TRGO//NCoSC). About 6 M aqueous KOH and Whatman filter paper were used as electrolyte and separator for HSC devices fabrication, respectively. Charge neutralization of the HSC device was carried out according to the Equation (3). The amount of the active electrode materials was calculated through mass balancing (Equation (4)) of both the electrodes. The electrochemical property of TRGO was measured using three-electrode configuration. The leaf like rectangular CV curves (Figure S6a) of TRGO indicated pure EDLC behavior. The specific capacitance of TRGO was determined from the GCD curve (Figure S6b). Although the specific capacitance of the TRGO was lower compared to the positive electrode material, it was used as negative electrode due to its higher potential window. The potential is an important parameter to improve the energy density of the supercapacitor device. The mass ratio of the positive and negative electrode (1:1.3) was optimized based on the GCD profiles. The total mass of the active electrode materials in HSC device was 1.4 mg. In three-electrode configuration, the working potential window of the positive electrode ranged from −0.1 to 0.4 V and for the negative electrode material it varied from −1 to 0 V. So, it is expected that the working potential window of the device should be equal to the sum of potential window of the positive and negative electrodes. In order to check the stable working potential window, CV was carried out in the working potential range of 1.2 to 1.55 V at a constant scan rate of 30 mV s$^{-1}$ (Figure 4b). The device was stable up to 1.5 V potential and electrolysis started beyond this range. Therefore, 0 to 1.5 V potential window was preferred for electrochemical performance study of the HSC device. The CV curves of the negative electrode, positive electrode, and HSC device are shown in Figure S6c. The CV curves of the HSC device at different scan rate from 10 to 100 mV s$^{-1}$ are shown in Figure 4c. The CV curves indicated that the HSC device showed Faradic behavior at higher voltage range and EDLC type behavior at lower voltage region. The shape of the CV curves remained unchanged at high scan rate of 100 mV s$^{-1}$, indicating good stability and rate performance of the electrode materials [1,14,18–20].

Resistance and charge transfer kinetics of the electrode materials of the device were investigated by EIS measurement. The impedance study of the device was performed from the Nyquist plot and was fitted by Z-View software with the help of equivalent Randles circuit model. The fitted EIS data (Figure 4d) demonstrated low Rs value (0.5 $\Omega$), suggesting good contact and well adhesion of the electrode materials with the current collector and good electrode/electrolyte contact at the interface. The device showed ~40 $\Omega$ $R_{ct}$ indicating faster charge transfer kinetics and good capacitive behavior of the device. The lower W-T (0.026) and W-P (0.16) values signified faster electrolyte ion diffusion of the electrodes in the device [5–7,31,38]. Figure 4e presents the GCD curves of the HSC device at different current densities from 2 to 15 A g$^{-1}$. The specific capacitance of the device was calculated from the discharge time of the GCD curve and the device showed high specific capacitance of ~142.6 F g$^{-1}$ at 2 A g$^{-1}$ current density. The device retained ~77.3% (110 F g$^{-1}$) of initial SCs and at 15 A g$^{-1}$ current density. The change in SCs and columbic efficiency with current density is shown in Figure 4f. It shows that the Columbic efficiency of the device increased with increasing the current density due to the dominating EDLC contribution toward the capacitive properties. The energy density (ED) and power density (PD) of the device was calculated using equation 5,6 and the device demonstrated

maximum ED of ~38.8 Wh Kg$^{-1}$ at PD of 1.3 kW Kg$^{-1}$. The highest PD of the device was 9.8 kW Kg$^{-1}$ at ED of ~30 Wh Kg$^{-1}$ (retained ~77.3%). The practical efficiency of the HSC device was demonstrated in Ragon plot (Figure 5a). This HSC device showed higher or comparative capacitive properties compared to the other devices as summarized in Table S1. Ultra-long-life cycle is an important criterion for a good supercapacitor device. Therefore, the stability of the fabricated HSC device was verified by performing the continuous GCD up to 10,000 cycles at constant current density of 12 A g$^{-1}$ as demonstrated in Figure 5b. The assembled HSC device showed high (~89.5%) retention in capacitance after 10,000 GCD cycles. The stability plot showed that the capacitance of the device increased sharply up to 1500 cycles and then decreased. This improvement in capacitance is mainly due to the perfect wetting of the electrode materials by electrolytes and ~100% activation of redox active sites. The shape of the GCD curve of 1st cycle and 10,000 cycles (Inset of Figure 5b) demonstrated that the shape of the GCD profile were not changed further suggesting good stability and reversibility of the device.

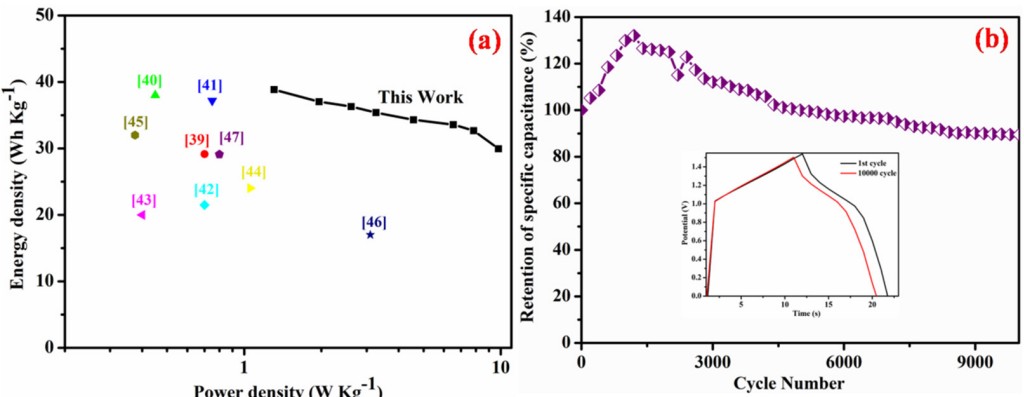

**Figure 5.** (**a**,**b**) Ragon plot and cyclic stability study of the device (inset of figure b is the GCD curve at 1st and 1000 cycles).

## 4. Conclusions

This study demonstrated a simple, environment-friendly, and cost-effective approach for the preparation of carbon-coated monometallic and bi-metallic transition metal sulfides as supercapacitor electrode materials. Herein, 3D monometallic carbon-coated NiS$_2$ (NiS$_2$@C) and bi-metallic nickel sulfide were synthesized by the introduction of Cu (NiCuS$_2$@C) and Co (NiCo$_2$S$_4$) separately. Effect of morphological and electrochemical properties by the introduction of external metals like Cu and Co on nickel sulfide as supercapacitor electrode was also investigated. The physicochemical and electrochemical studies demonstrated that the introduction of Cu and Co played an important role in tuning the morphology of the composites as well as electrochemical properties of the supercapacitor. In comparison to the NSC electrodes, the conductivity and redox active sites increased in the bimetallic NCuSC and NCoSC composites electrodes. The electrochemical experiment showed that the NCoSC electrode demonstrated higher capacitance of 760 F g$^{-1}$ at 2 A g$^{-1}$ current density as compared to the NCuSC and NSC electrodes. Therefore, a HSC device was fabricated by using NCoSC as the positive electrode and TRGO as the negative electrode. It showed highest energy density of ~38.8 Wh Kg$^{-1}$ at PD of 1.3 kW Kg$^{-1}$. When, the device exhibited highest power density of 9.8 kW Kg$^{-1}$, the device displayed minimum energy density of ~30 Wh Kg$^{-1}$ (retained ~77.3%). The HSC device also showed ~89.5% retention of specific capacitance after 10,000 cycles at a specific current of 12 A g$^{-1}$. So, the tuning of electronic and physical properties with the introduction of Cu and Co to nickel sulfide improved the supercapacitor performance.

**Supplementary Materials:** The following are available online at https://www.mdpi.com/article/10.3390/jcs5070180/s1. Figure S1: (a) FTIR spectra of GO and TRGO; (b) Raman spectra of TRGO.

Figure S2: (a,b) IFFT pattern of (A)and (B)regions, respectively of Figure 2a. Figure S3: FTIR spectra of NSC, NCuSC, and NCoSC electrodes. Figure S4: (a,b) CV curves of NSC and NCuSC electrodes at different scan rates, respectively. (c) Comparable discharge plots of NSC, NCuSC and NCoSC electrodes at 3 A g$^{-1}$ current density (d,e) GCD plots of NSC and NCuSC electrodes, respectively at different current density. Figure S5: (a–c) Z-View fitted EIS spectra of NSC, NCuSC, and NCoSC electrodes and (d) EIS spectra of NSC, NCuSC, and NCoSC at high frequency range. Figure S6: (a,b) CV and GCD plots of TRGO at different scan rates and current densities, respectively. (c) CV curves of positive, negative, and fabricated device in their respective potential window at constant 30 mV s$^{-1}$ scan rate. Table S1: Comparison of supercapacitor properties of some nickel-based hybrid supercapacitor (HSC).

**Author Contributions:** S.G.: conceptualization, methodology, formal analysis, investigation, writing—original draft. A.P. and P.S.: formal analysis, investigation, writing—review andediting. B.L. and S.K.M.: project administration, funding acquisition, validation. S.S. and G.G.D.: project administration, resources. N.C.M.: project administration, software, validation. T.K.: supervision, formal analysis, investigation, writing—review and editing. All authors have read and agreed to the published version of the manuscript.

**Funding:** This research was funded by Defence Research and Development Organization (DRDO), Ministry of Defence, Government of India for the financial support [DFTM/02/3111/M/01/JCBCAT/1288/D(R&D), Dated: 07/07/2017] through the GAP219012 project.

**Acknowledgments:** The authors are thankful to the Director of CSIR-CMERI, Durgapur. Authors are also thankful to JCBCAT, Kolkata.

**Conflicts of Interest:** The authors declare no conflict of interest.

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
