# Peer review of "Influence of Transition Metals (Cu and Co) on the Carbon-Coated Nickel Sulfide Used as Positive Electrode Material in Hybrid Supercapacitor Device"

_jcs, doi:10.3390/jcs5070180_

Round 1
Reviewer 1 Report
Many data, including material characterization, electrochemical analysis, and device performance were reported in this work by Ghosh et. al. However, this reviewer feels some of the data presented do not well support the analysis and conclusions drawn in the manuscript. Some of the claims have fatal flaws.
- Page 2. “ However, many transition metals such as Ru, Mn, Ni, etc. exhibit wide band gap semiconductors. Introducing good semiconductor transition metal atoms (such as Cu and Co) together with any transition metal-based electrode material…”. The concept of metals cannot be mixed up with that of semiconductors.
- 1a. For NCuSC sample, if all XRD peaks are assigned to either NiS2 or CuS, then this material is not an alloy but a simple mixture of NiS2 and CuS. Do the two different particles in Fig. 1c represent NiS2 and CuS, respectively?
- A better image should be used for Fig. 1b. It’s obvious that Fig. 1d is elongated in the lateral direction, which is not acceptable.
- 2a seems to be irrelevant.
- 2c. The bandgap fitting seems to be too arbitrary to be true.
- 3a. “The CV curves also suggested that the NCoSC and NCuSC electrodes showed pure battery like behaviour and the NSC electrodes showed pseudocapacitive like charge storage behavior…”. It’s hard to understand how such a conclusion was drawn, further considering that in Fig. 3c, the fitted “b” values are in the range of 0.63 to 0.68.
- In Table 1, what are the meanings of “-T”, “-P”?
- There is no inset in Fig. 5b.
- There are no values for the Z’ axis in Fig. S2d.
Author Response
Journal: Journal of Composite Science
[J. Compos. Sci.] Manuscript ID: jcs-1289119
Title: “Influence of transition metals (Cu and Co) on the carbon-coated nickel sulfide used as positive electrode material in hybrid supercapacitor device"
Dear Editor
I am sending herewith the revised manuscript (jcs-1289119) entitled “Influence of transition metals (Cu and Co) on the carbon-coated nickel sulfide used as positive electrode material in hybrid supercapacitor device” for the publication in the journal “Journal of Composite Science”. We are thankful to the Editor and Reviewers for their valuable comments which make our paper more informative. Below are our answers to the reviewer’s questions. The changes made in the revised manuscript have been highlighted in red color.
We are thankful to the Editor for kindly considering our manuscript. We are also grateful to the reviewers for their valuable comments on our work. The comments were helpful for the up gradation of the manuscript. We have addressed all the comments raised by the reviewer and modified the manuscript accordingly.
#Reviewer 1
Many data, including material characterization, electrochemical analysis, and device performance were reported in this work by Ghosh et. al. However, this reviewer feels some of the data presented do not well support the analysis and conclusions drawn in the manuscript. Some of the claims have fatal flaws.
We are thankful to you for kindly evaluating the manuscript.
Q1. Page 2. “However, many transition metals such as Ru, Mn, Ni, etc. exhibit wide band gap semiconductors. Introducing good semiconductor transition metal atoms (such as Cu and Co) together with any transition metal-based electrode material…”. The concept of metals cannot be mixed up with that of semiconductors.
A1. We are thankful to you for kindly evaluating our manuscript. We are fully agreed with your observation. The sentence has been deleted in the revised manuscript.
Q2. 1a. For NCuSC sample, if all XRD peaks are assigned to either NiS2 or CuS, then this material is not an alloy but a simple mixture of NiS2 and CuS. Do the two different particles in Fig. 1c represent NiS2 and CuS, respectively?
A2. All the PXRD peaks related to CuS and NiS2 particles were present in the NCuSC electrode. However, the positions of the peaks slightly shifted in the NCuSC as compared to the peaks of CuS and NiS2 particles. These observations suggested chemical interaction between the CuS and NiS2 particles and formation of NiCuS2 composite materials. On the other hand, the FE-SEM image of NCuSC electrode (Figure 1c) also showed the appearance of two different types of particles of CuS and NiS2 particles. The FE-SEM image of NCoSC composite electrode (Figure 1d) exhibited unique size, well distribution and interconnected particles. This type of interconnected network of the particles may increase the number of electrochemical active sites and facilitate the good electrical conductivity and enhance the electrochemical properties of the supercapacitor device [24-25]. Therefore, the FE-SEM images showed that the introduction of another transition metals (Cu and Co) with NiS2@C, resulted uniform growth of NCuSC and NCoSC particles with smaller in size, which may increase the active space for electrolytes and enhance the diffusion of the electrolytes ion.
Q3. A better image should be used for Fig. 1b. It’s obvious that Fig. 1d is elongated in the lateral direction, which is not acceptable.
A3. We are very thankful for kindly evaluating the manuscript. The manuscript has been revised accordingly.
Q4. 2a seems to be irrelevant
A4. The FT-IR spectra shown in Figure 2a represent the presence of oxygen groups in the coated carbon. The peak at 655 cm-1 is possibly due to the Ni-S-Ni bending vibration. NCuSC and NCoSC electrodes showed another peak at 784 cm-1 due to the stretching vibration of Cu-S/Co-S. Therefore, the FT-IR-spectra validate the formation of NCoSC nd NCuSC electrodes. Based on your suggestion, Figure 2a has been moved from main text to the supporting information.
Q5. 2c. The bandgap fitting seems to be too arbitrary to be true.
A5. The UV-vis spectra and Tauc’s plots are re-investigated. The spectrum of all the electrodes (Figure 2b) showed two peaks at different regions of 210 to 230 nm and 260 to 280 nm. A broad absorbance band at ~275 nm region of NSC electrode implies the presence of metal sulphides. This broad peak slightly shifted to lower wavelength region for NCuSC and NCoSC electrodes (~260 nm) due to the introduction of Cu and Co in carbon coated nickel sulphides compounds [29]. Another peak appeared at ~ 224 nm region of NCuSC and NCoSC electrodes correspond to the stretching vibration of –C=O groups of coated carbon. This peak slightly shifted to the lower wavelength of ~214 nm in NSC electrode. The Tauc’s plots were also re investigated (Figure 2c) and the band gap energy was calculated from the Tauc’s plot. The plots demonstrate that the band gap energy of NCS (~3.1 eV) electrode was higher as compared to the NCuSC (~2.92 eV) and NCoSC (~2.76 eV) electrodes.
Q6. 3a. “The CV curves also suggested that the NCoSC and NCuSC electrodes showed pure battery like behavior and the NSC electrodes showed pseudocapacitive like charge storage behaviour…”. It’s hard to understand how such a conclusion was drawn, further considering that in Fig. 3c, the fitted “b” values are in the range of 0.63 to 0.68.
A6. The comparable CV curves of NSC, NCuSC and NCoSC electrodes shown in Figure 3a demonstrated that the nature of the CV curves for NSC electrode is different than the NCuSC and NCoSC electrodes. The CV curves represent intense anodic and cathodic redox peaks for NCuSC and NCoSC electrodes as compared to the broad hump appeared in NSC electrode. This observation suggested that the NCoSC and NCuSC electrodes showed pure battery like behavior. In contrary, the capacitive property of NSC electrode was governed by the battery like charge storage process as well as the surface controlled pseudocapacitive charge storage mechanism. The larger ‘b’ value of NSC electrode (0.68) compared to the NCuSC (0.64) and NCoSC (0.63) also implied more surface controlled capacitive charge storage process.
Q7. In Table 1, what are the meanings of “-T”, “-P”?
A7. The CPE can be divided into two physical constituents, one is the phase element exponent (CPE-P) which means the nature of the charge transfer process in the high frequency region and the other is the capacitance extent (CPE-T) that characterizes the double layer capacitance of active electrode materials. On the contrary, the Warburg element (Wo) can also be divided into three components of Warburg resistance (W-R), diffusion time constant (W-T) and the exponent (W-P). W-T represents the how fast electrolyte ions are diffuse and W-P represents the capacitive diffusion nature of the electrolytes.
Q8. There is no inset in Fig. 5b.
A8. Fig. 5b has been modified accordingly in the revised manuscript.
Q9. There are no values for the Z’ axis in Fig. S2d
A9. We are sorry for that mistake and the Fig. S2d has been modified accordingly in the revised manuscript.

Reviewer 2 Report
Abstract:
- The authors wrongly stated Copper as Co and Cobalt as Cu. Please, correct.
- please define HSC and TRGO.
Introduction:
Stating the importance of the materials used in their study can not work as a motivation for the study. The authors need to provide information on the gap in the literature and how their research would add value to the scientific community.
Characterization
1. The authors claimed they have synthesized carbon-coated Ni-S electrodes. However, their characterization did not prove such a claim. The authors should provide any of the following or all of them if necessary: Cross-section SEM, HR-TEM, and In-depth XPS.
2. The authors should provide XPS for all samples to prove the presence of the claimed transition states and to provide information on the exact ratio of the elements in their samples.
Electrochemistry
- The authors should provide an enlarged image for the low-frequency region in the Nyquist plot.
- The authors should explain their low Coulombic efficiency in low current density.
- The authors should provide a comparative table comparing their results with previously published results on the same electrode materials they used to show the enhancement they did to the field.
Author Response
Journal: Journal of Composite Science
[J. Compos. Sci.] Manuscript ID: jcs-1289119
Title: “Influence of transition metals (Cu and Co) on the carbon-coated nickel sulfide used as positive electrode material in hybrid supercapacitor device"
Dear Editor
I am sending herewith the revised manuscript (jcs-1289119) entitled “Influence of transition metals (Cu and Co) on the carbon-coated nickel sulfide used as positive electrode material in hybrid supercapacitor device” for the publication in the journal “Journal of Composite Science”. We are thankful to the Editor and Reviewers for their valuable comments which make our paper more informative. Below are our answers to the reviewer’s questions. The changes made in the revised manuscript have been highlighted in red color.
We are thankful to the Editor for kindly considering our manuscript. We are also grateful to the reviewers for their valuable comments on our work. The comments were helpful for the up gradation of the manuscript. We have addressed all the comments raised by the reviewer and modified the manuscript accordingly.
#Reviewer 2
Q1. The authors wrongly stated Copper as Co and Cobalt as Cu. Please, correct.
A1. We are sorry for this mistake and the manuscript has been revised accordingly.
Q2. Please define HSC and TRGO.
A2. In the abstract section and main text, we have defined the HSC (Hybrid supercapacitor) and TRGO (Thermally reduced graphene oxide) in the revised manuscript.
Introduction:
- Stating the importance of the materials used in their study cannot work as a motivation for the study. The authors need to provide information on the gap in the literature and how their research would add value to the scientific community.
- Lower electronegativity of the sulphur as compared to the oxides facilitates the electron transfer process and improves the conductivity of the electrode materials. However, the agglomeration tendency of nickel sulphide-based electrode materials reduces electrochemically active sites, resulting in poor electrochemical properties like inferior rate capability and stability. On the other hand, carbonaceous materials reduce the agglomeration of the TMC-based materials resulting larger electroactive surface area and improved electrical conductivity. In an another approach, a thin layer coating of carbon on the TMC materials further increase the structural stability of the active electrode materials during long charge-discharge cycles. A few literatures showed the use of only nickel/copper/cobalt chalcogenide-based or nickel chalcogenide-based bi-metallic composite as electrode materials in supercapacitor devices [16-18]. Tavakoli et al. reported the use of yolk-shell like CuCo2Se4 microspheres as supercapacitor electrode material. The hollow yolk-shell like structure decreased the mass/charge transfer distance and increased the rate capability with power density of the supercapacitor. It showed high specific capacitance of ~512 F g-1 at 1 A g-1 current density [1]. Ma et al. showed that the intertwined fiber structure is advantageous for supercapacitor applications. It was found that the NiCo2S4 embedded carbon fiber displayed improved capacitive properties compared to the pure NiCo2S4 [5]. X. sun and co-workers prepared CNT@NiMn2O4 core-shell nano-structured composite through microwave assisted hydrothermal techniques [8]. In comparison to the pure NiMn2O4, the CNT@NiMn2O4 core-shell structure exhibited better supercapacitor performance due to the improved electrolytes transportation. Wan et al. synthesized anisotropic carbon from the wood derivative and formed a composite with copper oxide for supercapacitor electrode material. The abundant anisotropic channels present in carbon facilitated rapid transportation of electrolytes and reduced the resistance of the electrode material. The electrode material exhibited maximum specific capacitance of 694.8 F g-1 at 0.5 A g-1 current density [9]. Therefore, it is seen that the porous carbon atoms can reduce the resistance and increase the rate capability and power density of the supercapacitor device. However, the effect of the addition of transition metal atoms in nickel chalcogenide-based electrodes on the morphology, electrical properties and charge storage performance has not been reported so far.
Based on the above considerations, carbon-coated bi-metallic nickel sulphide-based composite was prepared by one-step hydrothermal technique for the development of high-performance electrode materials for supercapacitor. In this study, introduction of transition metals (Co and Cu) into the carbon coated nickel sulphide is expected to improve the performance of the supercapacitor electrode. The influence of the physical and electrochemical properties of electrode materials for the interrogation of two transition metals separately was systematically studied.
Characterization:
Q1. The authors claimed they have synthesized carbon-coated Ni-S electrodes. However, their characterization did not prove such a claim. The authors should provide any of the following or all of them if necessary: Cross-section SEM, HR-TEM, and In-depth XPS.
A1. We are thankful to you for kindly evaluating the manuscript. The HR-TEM image of NCoSC electrode was carried out to characterize the carbon coated nickel sulphide electrodes. The HR-TEM image of NCoSC electrode (Figure 2a) showed the network like interconnected structure of the particles. The d-spacing of the NCoSC electrode was calculated in the A and B region as shown in Figure 2a. The fast Fourier transformation (FFT) followed by inverse FFT (IFFT) was carried out in the selected area. The d-spacing obtained from the IFFT images from region A (Figure S2a) was found to be 0.23 nm confirming the presence of (400) plane of NiCo2S4. The d-spacing of 0.366 nm in region B indicated the (002) plane of coated carbon. The incomplete destruction of the crystallographic features of (002) plane of coated carbon suggested the presence of amorphous nature of the coated carbons. The HR-TEM images confirmed the carbon coating on the NiCo2S4 particles and formation of NCoSC composite electrode materials. The (400) plane of NiCo2S4 and (002) plane of carbon as evidenced from the HR-TEM image analysis matched well with the PXRD pattern of NCoSC.
We are agreed with the reviewer that the XPS is very helpful in determining the chemical state of an element present in a material. We are extremely sorry to inform you that we don’t have this facility in our Institute and we are always outsourcing it from the other Academic/R&D Organizations of the Country. However, due to the COVID-19 pandemic most of the Academic/R&D Organizations of the Country [INDIA] are partly open or closed completely right now. Therefore, it is very difficult for us to provide the XPS analysis results of our samples at this moment. However, we have tried to establish the structure & morphology of the prepared samples by other characterization techniques.
Q2. The authors should provide XPS for all samples to prove the presence of the claimed transition states and to provide information on the exact ratio of the elements in their samples.
A2. We are very thankful to you for your valuable suggestion. We are agreed with the reviewer that the XPS is very helpful in determining the chemical state of an element present in a material. We are extremely sorry to inform you that we don’t have this facility in our Institute and we are always outsourcing it from the other Academic/R&D Organizations of the Country. However, due to the COVID-19 pandemic most of the Academic/R&D Organizations of the Country [INDIA] are partly open or closed completely right now. Therefore, it is very difficult for us to provide the XPS analysis results of our samples at this moment. However, we have tried to establish the structure & morphology of the prepared samples by other characterization techniques. I hope that you will be satisfied with the revised version of the manuscript.
Electrochemistry
Q1. The authors should provide an enlarged image for the low-frequency region in the Nyquist plot.
A1. The enlarged image of the Nyquist plot is providing in the Figure S5d, in the revised manuscript.
Q2. The authors should explain their low Coulombic efficiency in low current density.
A2. The columbic efficiency of the electrode materials were calculated from the discharge time to charge time ratio. The main charge storage in NCoSC occurred through the slow irreversible faradic process at lower current density. However, at higher current density the charge storage mechanism is governed by the fast double layer type capacitive process. Therefore, at high current density the charge storage process is shifted to double layer type capacitive process from the slow irreversible faradic process and the columbic efficiency is increased with the sacrifice of specific capacitance.
Q3. The authors should provide a comparative table comparing their results with previously published results on the same electrode materials they used to show the enhancement they did to the field.
A3. The capacitive properties of the fabricated device has been compared to the other devices using nickel based electrode materials as summarized in Table S1 of the revised manuscript.

Reviewer 3 Report
Reviewer’s comments:
Manuscript ID: JCS-1289119
The authors demonstrated the effect of morphology and electrochemical performance of carbon-coated nickel sulfide composite electrodes by introducing copper (Co) and cobalt (Cu). The electrochemical experiment shows that the NCoSC electrode exhibits the highest capacitance value of ~760 F g-1 at 2 A g-1 current density compared to the NCuSC and NSC electrodes. Most impressively, the fabricated device demonstrates a maximum energy density of ~38.8 Wh Kg-1 and a power density of 9.8 kW Kg-1. Furthermore, the HSC device also shows ~89.5% retention in specific capacitance after 10,000 charge-discharge cycles at 12 A g-1 current density. Overall, it is an interesting and novel work and is appropriate for publication in the J. Compos. Sci. Therefore, I recommend its acceptance for publication in Chemistry of Materials Journal after the following minor revisions:
Questions and comments:
- Authors should provide the Raman spectra of all the samples for characterizing the nature of coated carbon.
- To prepared asymmetric devices, authors used TRGO as a negative electrode material, but no data is provided to confirm the reduction of GO to TRGO. So, the authors should provide some data to ensure the reduction process.
- Fig. 5a: In the Ragon plot, the data from recent literature for devices can be included to compare the state-of-art of the present work.
- The authors should make a comparison table of electrochemical property of recently reported nickel-based electrode materials.
- The title can be modified as “Influence of transition metals (Cu and Co) on the carbon-coated nickel sulfide used as positive electrode material in hybrid supercapacitor device.”
Author Response
Journal: Journal of Composite Science
[J. Compos. Sci.] Manuscript ID: jcs-1289119
Title: “Influence of transition metals (Cu and Co) on the carbon-coated nickel sulfide used as positive electrode material in hybrid supercapacitor device"
Dear Editor
I am sending herewith the revised manuscript (jcs-1289119) entitled “Influence of transition metals (Cu and Co) on the carbon-coated nickel sulfide used as positive electrode material in hybrid supercapacitor device” for the publication in the journal “Journal of Composite Science”. We are thankful to the Editor and Reviewers for their valuable comments which make our paper more informative. Below are our answers to the reviewer’s questions. The changes made in the revised manuscript have been highlighted in red color.
We are thankful to the Editor for kindly considering our manuscript. We are also grateful to the reviewers for their valuable comments on our work. The comments were helpful for the up gradation of the manuscript. We have addressed all the comments raised by the reviewer and modified the manuscript accordingly.
#Reviewer 3
The authors demonstrated the effect of morphology and electrochemical performance of carbon-coated nickel sulfide composite electrodes by introducing copper (Co) and cobalt (Cu). The electrochemical experiment shows that the NCoSC electrode exhibits the highest capacitance value of ~760 F g-1 at 2 A g-1 current density compared to the NCuSC and NSC electrodes. Most impressively, the fabricated device demonstrates a maximum energy density of ~38.8 Wh Kg-1 and a power density of 9.8 kW Kg-1. Furthermore, the HSC device also shows ~89.5% retention in specific capacitance after 10,000 charge-discharge cycles at 12 A g-1 current density. Overall, it is an interesting and novel work and is appropriate for publication in the J. Compos. Sci. Therefore, I recommend its acceptance for publication in Chemistry of Materials Journal after the following minor revisions:
We are thankful to you for kindly evaluating the manuscript. We have modified the manuscript as per the reviewers’ comments. We have modified the sentences in the revised manuscript. I hope that you will find everything in order in the revised manuscript. We are again thankful to the Reviewer for appreciating our work. The work has been submitted based on the invitation for the special issue and we will be grateful to you if you are kindly satisfied with the revised version of the manuscript.
Q1. Authors should provide the Raman spectra of all the samples for characterizing the nature of coated carbon.
A1. The Raman spectra of NSC, NCuSC and NCoSC electrode materials are shown in Figure 2a. The Raman peaks at 1356 and 1578 cm-1 correspond to the D and G modes respectively. The higher G band intensity compared to the D band in all the composite materials suggested that the coated carbon atoms were more sp2 hybridized and the disorder nature of the carbon atoms were smaller. The ID/IG ratio decreased from NSC (0.81) to NCuSC (0.70) to NCoSC (0.65) electrodes periodically implying the decrease in disorder structure of coated carbon and the synergistic effect of metal sulphide with carbon. A broad 2D band present in all the electrode materials at 2850 cm-1 region was ascribed to the graphitic nature of the coated carbon. The 2D band intensity increased greadually from NSC to NCuSC to NCoSC electrode. It is suggested that the coated carbon on metal sulfide formed a layered structure similar to graphite
Q2. To prepared asymmetric devices, authors used TRGO as a negative electrode material, but no data is provided to confirm the reduction of GO to TRGO. So, the authors should provide some data to ensure the reduction process.
A2. The formation of TRGO was confirmed by FT-IR and Raman spectra. The FT-IR spectra shown in Figure S1a demonstrated that the peak intensity of oxygenated groups at 1622, 1400, 1118 and 1030 cm-1 region decreased remarkably after the thermal treatment of GO. Raman spectra study (Figure S1b) also showed higher G band (1360 cm-1) intensity compared to D band (1360 cm-1) confirming the lower disorder in TRGO and larger number of sp2 hybridized carbon atoms. Appearance of a sharp 2D band at ~2765 cm-1 region confirmed the formation of a few layers TRGO.
Q3. Fig. 5a: In the Ragon plot, the data from recent literature for devices can be included to compare the state-of-art of the present work.
A3. Thank you for kind evaluating our manuscript. The manuscript has been revised accordingly.
Q4. The authors should make a comparison table of electrochemical property of recently reported nickel-based electrode materials.
A4. As per your suggestion a comparison table of this work with recently reported nickel based supercapacitor device is making and provide in the supporting information (Table S1) in the revised manuscript.
Q5. The title can be modified as “Influence of transition metals (Cu and Co) on the carbon-coated nickel sulfide used as positive electrode material in hybrid supercapacitor device.”
A5. We are thankful to you for kindly evaluating the manuscript. Title of the manuscript has been revised accordingly.

Round 2
Reviewer 1 Report
None
Reviewer 2 Report
The authors considered the required edits. I suggest accepting the paper in its current form.